# An Algorithm Based on Text Position Correction and Encoder-Decoder Network for Text Recognition in the Scene Image of Visual Sensors

**DOI:** 10.3390/s20102942

**Published:** 2020-05-22

**Authors:** Zhiwei Huang, Jinzhao Lin, Hongzhi Yang, Huiqian Wang, Tong Bai, Qinghui Liu, Yu Pang

**Affiliations:** 1School of Communication and Information Engineering, Chongqing University of Posts and Telecommunications, Chongqing 400065, China; hzwnet@swmu.edu.cn; 2School of Medical Information and Engineering, Southwest Medical University, Luzhou 646000, China; 3Chongqing Key Laboratory of Photoelectronic Information Sensing and Transmitting Technology, Chongqing University of Posts and Telecommunications, Chongqing 400065, China; yhz5256@163.com (H.Y.); wanghq@cqupt.edu.cn (H.W.); baitong03@126.com (T.B.); lqh106@163.com (Q.L.)

**Keywords:** scene text recognition, visual sensor, text position correction, encoder-decoder network

## Abstract

Text recognition in natural scene images has always been a hot topic in the field of document-image related visual sensors. The previous literature mostly solved the problem of horizontal text recognition, but the text in the natural scene is usually inclined and irregular, and there are many unsolved problems. For this reason, we propose a scene text recognition algorithm based on a text position correction (TPC) module and an encoder-decoder network (EDN) module. Firstly, the slanted text is modified into horizontal text through the TPC module, and then the content of horizontal text is accurately identified through the EDN module. Experiments on the standard data set show that the algorithm can recognize many kinds of irregular text and get better results. Ablation studies show that the proposed two network modules can enhance the accuracy of irregular scene text recognition.

## 1. Introduction

The object of natural scene text recognition is to identify the text in the image of natural scene. Natural scene text recognition has important applications in intelligent image retrieval [1,2], license plate recognition [3], automatic driving [4], scene image translation [5] and many other fields.

In recent years, although many effective text recognition methods [6,7,8,9,10,11,12] have been proposed and the performance of text recognition has been greatly improved, the text recognition technology of natural scene still has some shortcomings. For the text of natural scene, there is a variety of permutation directions between adjacent texts. In addition to the linear permutation, they may also be arranged in irregular directions such as arcs [13]. For natural scene text arranged in multiple directions, the bounding box may be a rotating rectangle or quadrilateral, so it is difficult to design an effective method to calculate the regularity of the direction of arrangement between adjacent texts [14]. In addition, the irregularity of the visual features of the deformed scene text also hinders the further development of the text recognition technology [15].

The wide variety of text and the diversity of the spatial structure of different types make the visual characteristics of text area have great differences [16], so it is difficult to find a good description feature to classify text area and background area. Therefore, it is also a difficult work to build a multi-classification text recognition framework. Further research is still needed to reach the practical level.

For this reason, we propose a text recognition algorithm based on TPC-EDN to realize a better recognition of various types of irregular text in natural scenes. The algorithm uses TPC module to modify the slanted text into horizontal text for easy recognition, and then accurately identifies the text content through EDN model. The encoder network (EN) module uses dense connection network and BLSTM to effectively extract the spatial and sequence characteristics of text and generate coding vectors. The decoder network (DN) module converts the encoding vector into the output sequence through the attention mechanism and LSTM.

Our contributions in this paper are as follows: First, we propose a TPC approach which is a coordinate offset and regression method based on CNN to realize the end-to-end training. Second, we introduce EN module to extract text features based on dense connection network and BLSTM. Third, the training process of our proposed algorithm is simple and fast, and it is robust to irregular text recognition.

## 2. Overall Network Structure

The text recognition algorithm designed in this paper mainly includes two modules: the text position correction module and the encoder-decoder network module. The TPC module corrects the detected oblique text into horizontal text, then the EDN module recognizes horizontal text. EDN module includes the encoder network (EN) and the decoder network (DN). The EN uses the dense block and two-layer BLSTM [17] methods to extract text features, and can generate feature vector sequences with character context feature relations. The DN uses the attention mechanism [18] to weight the encoded feature vectors, which can make more accurate use of character-related information. Then, through a layer of LSTM [19], DN adopts the output of the previous moment and the input of the current moment to jointly determine the recognition result of the current moment. The overall structure is shown in Figure 1.

### 2.1. Text Position Correction Module

TPC is the main research method for the oblique text recognition, which corrects the oblique text into the horizontal text, and then carries on the recognition to the horizontal text. Most of the traditional text position correction methods are based on affine transformation [20], which has good effects on text with small tilt angle, but bad effects on text with large tilt angle and are difficult to train. In the study of text recognition algorithm, this paper proposes an improved TPC method based on the idea of variable convolution two-dimensional offset [21] and offset sampling [22], which is a coordinate offset regression method based on CNN. It can be combined with other neural networks to complete end-to-end training, and the training process is simple and fast. The detailed structure is shown in Figure 2.

As can be seen from Figure 2, the TPC process of this paper is as follows: Firstly, a pre-processing step is carried out to process the input text into the same size, which can speed up the training process of the algorithm. Secondly, the spatial features of pixels [23] are extracted by CNN to obtain a fixed size feature map, in which each pixel corresponds to a part of the original image. This is equivalent to splitting the original image into several small pieces, and the prediction of coordinate offset for each piece is the same as the two-dimensional offset prediction of the deformable convolution. Thirdly, the offset is then superimposed on the normalized coordinates of the original image. Finally, the Resize module uses a bilinear interpolation method to sample the text feature map to the original size as the revised text.

The input of the whole text recognition algorithm is the text bounding box detected by the text detection algorithm. Due to the irregular shape of the text, the size of the detected text bounding box is different. If the text is directly input into the text recognition algorithm, the training speed of the text recognition algorithm will be reduced. Therefore, after the preprocessing module, the text bounding box is fixed to a uniform size, namely 64 pixels in height and 200 pixels in width, and then the feature map is obtained by extracting features continuously through CNN, and the coordinate offset is returned. The detailed structure and parameter configuration of TPC are shown in Table 1.

In Table 1, k3 means the size of convolution kernel is 3 × 3, num64 means the number of convolution kernel is 64, s1 means the stride is 1, p1 means the padding is 1, Conv means convolution, and AVGPool means average pooling. The number of convolution kernels gradually increases from the first layer, and then decreases. Finally, the number is set as 2 in order to generate a two-dimensional offset feature map, whose size is 2 × 11. This is equivalent to dividing the entire input image into 22 blocks, each corresponding to the corresponding coordinate offset value. The activation function Tanh is used to adjust the predicted value of the migration to between [−1, 1], and return the offset of the *X*-axis and the offset of the *Y*-axis through two channels respectively. Then, the Resize module is used to sample the offset feature map of the two channels to the size of the original figure 2 × 64 × 200. Sample is a bilinear interpolation up-sampling module to obtain the revised text.

Each value in the offset feature map represents its corresponding coordinate offset of the point in the original image. In order to correspond to the dimension of the feature map, the coordinates of each pixel in the original image need to be normalized. The normalized coordinate interval is between [−1, 1], and it also contains two channels, namely the *X*-axis channel and *Y*-axis channel. Figure 3 is the comparison of the original image before and after the normalization of coordinates.

The image is stored in the form of matrix in the computer, so the upper left corner of the image in Figure 3 is the origin of the coordinate axis (0,0), the horizontal axis represents the width of the image, and the vertical axis represents the height of the image. After normalization, the center of the image is the origin of the coordinate, the upper left corner in Figure 3 is the coordinate (−1,−1), and the lower right corner is the coordinate (1,1). The generated normalized image is double-channel, and the coordinates of pixels in the same position on different channels are the same. After that, the offset feature image is superimposed with the corresponding area of the normalized image to complete the correction of the corresponding position of each pixel. The formula is expressed as:(1)$$T_{\left(channel,i,j\right)}=offset_{\left(channel,i,j\right)}+G_{\left(channel,i,j\right)},channel=1,2$$
(2)$$F_{{}_{\left(ii^{\prime },jj^{\prime }\right)}}^{\prime }=F(i^{\prime },j^{\prime })$$
where, *channel* refers to the number of channels, *T* represents the feature map after position correction, *offset* represents the offset feature map, *G* represents the normalized image, (*i*, *j*) represents the coordinates of the normalized image, (*i’*, *j’*) represents the coordinates of the original image, (*ii’*, *jj’*) represents the revised offset coordinates, *F’* represents the corrected image, *F* represents the original image.

Adding the corresponding offset to the normalized image, the offset of each point coordinate on the normalized image occurs in both horizontal and vertical directions. The offset is (∆x, ∆y), the revised offset coordinate is (*ii*, *jj*), and then the size is up-sampled to the original size by bilinear interpolation method. The revised image *F’* is obtained, whose corresponding coordinate is (*ii’*, *jj’*), The relation between the original image and the normalized image is shown in Formula (2). The pixels of the two points remain the same size, just the position coordinates are changed.

### 2.2. Encoder Network

The EN module encodes the spatial and sequential features of extracted text images into fixed feature vectors [24]. Feature extraction network [25] plays a key role in the EN module. A good feature extraction network can determine the quality of encoding and has a great impact on the recognition effect of the whole text recognition algorithm. In this paper, the EN module adopts the methods of dense connection network and BLSTM to extract text features, in which dense connection network can extract rich spatial features of text images. Considering the context sequence feature of text, the feature relation between different characters can be learned by BLSTM. The EN module designed in this paper is easy to train and has a good effect. A brief introduction about it is as follows:

(1) Dense connection network is stacked by several dense blocks. Taking the advantages of DenseNet [26] in feature extraction, dense connection network is used to improve the direction of information flow during feature extraction, and all layers in a dense block can be connected by jumping. Each convolution layer can obtain feature information from all previous layers, enhance the reuse of multi-layer features, and transmit feature information to all subsequent layers. At the same time, the method of jumping direct connection makes it easier to obtain the gradient in the process of back propagation, simplifies the feature learning process, and alleviates the gradient dispersion problem.

(2) The detailed structure of the two BLSTMs is shown in Figure 4. Each BLSTM has two hidden layers, recording two states of the current time t: one is the forward state from front to back, the other is the reverse state from back to front. The input of the first layer is the sequence of feature vectors extracted by CNN {x_0_, x_1_, …, x_i_}, the output after a layer of BLSTM is {$y_{0}^{1}$, $y_{1}^{1}$, …, $y_{i}^{1}$}. And then it is taken as the input of the second layer, finally the output sequence {$y_{0}^{2}$, $y_{1}^{2}$, …, $y_{i}^{2}$} can be got. As can be seen from Figure 4, the output of each time t is determined by the hidden layer state in both directions. In this paper, two BLSTMs are stacked to learn the feature states of the four hidden layers, which can not only store more memory information, but also better learn the relationship between feature vectors.

The dense block generates a two-dimensional feature map, while the input of BLSTM is in the serialized form. Therefore, it is necessary to convert the feature map into the sequence form of feature vectors, and then learn the context feature relationship between sequences through BLSTM. Figure 5 shows the process of transforming the feature map into the feature vector sequence. The feature map is evaluated according to the column of a certain width, the vertical direction is taken as a feature vector.

As can be seen from Figure 5, the character “O” requires multiple feature vectors to determine the output value, and it is impossible to accurately predict the character by relying on only one feature vector. Therefore, learning the correlation between feature vectors through BLSTM plays an important role in character recognition.

The EN module adopts four dense blocks, followed by two convolution layers, between which there is one Max Pooling and activation function layer, and then two BLSTM layers. The detailed parameters of the EN module are shown in Table 2.

As can be seen from Table 2, EN module adopts several convolution layers, pooling layers and activation function layers. The detailed parameters of the convolution layer include the size of convolution kernel, the number of convolution nuclei, stride and padding, which are respectively represented by k, num, s and p. The Max Pooling method is adopted in the all pooling layers, and the parameters are convolution kernel size k and stride s. The activation function takes the Swish function. The number of convolution nuclei in the four dense blocks gradually increases. In each Dense Block, “×4” represents four consecutive convolution layers, followed by two convolution operations. Finally, two BLSTMs are adopted, in which the number of hidden layer units of BLSTM in each layer is 256.

### 2.3. Decoder Network

The DN module is the reverse process of the EN module, which decodes the encoded feature vectors into output sequences and makes the decoding state as close as possible to the original input state. The text area of a text image usually exists in the form of a sequence, with variable length, and its feature vector is serialized. Therefore, this paper adopts soft attention mechanism [27] to focus the serialized feature vectors according to the weight distribution, which can effectively use the character features at different moments to predict the output value, and finally connects a layer of LSTM, which can store the past state and determine the output of the current moment through the output of the previous moment. The detail structure on DN is shown in Figure 6.

Figure 6 shows that the feature vector sequence generated by the EN is directly used as the input of the DN, the hidden layer of BLSTM in the process of the EN contains context feature of text feature vector sequence, the feature vector set can be set as [h_1_, h_2_, …, h_i_, …, h_T_], in which the feature Hi generated at each moment i consists of two directions of feature combination, h_i_ = [h_i_, $h_{i}^{*}$]. *C_t_* is the semantic encoding vector of the attention model, represents the weighted value of hidden layer feature hi at time t in LSTM network, is expressed as Equation (3).

In Figure 6, T represents the attention range of the attention mechanism, and its length is 30. If T is too large, the hidden layer needs to remember too much information, the calculation of the model increases rapidly, and the general text statement rarely exceeds 30 words. And too large T value will also make the model’s attention be distracted, so that the DN module cannot focus on the key feature vectors, and the decoding effect is not good. In this paper, the designed DN module takes the predicted output of the previous moment as the input of the current moment through LSTM, which can serve as a reference for the prediction of the current moment. In Figure 6, the output at the current moment can be accurately determined to be “P” based on the past output state. The detailed Formula (3)–(7) of the whole decoding process is as follows:(3)$$C_{t}=\sum_{i=1}^{T}A_{t,i}h_{i}$$
(4)$$A_{t,i}=\frac{\exp(e_{t,i})}{\sum_{k=1}^{T}\exp(e_{t,k})}$$
(5)$$e_{t,i}=f_{att}(s_{t-1},h_{i})$$
(6)$$s_{t}=f(s_{t-1},y_{t-1},C_{t})$$
(7)$$y_{t}=g(y_{t-1},s_{t},C_{t})$$

In the above Equations (3)–(7), *A*_t,_*_i_* represents the attention weight after normalization, *e_t,i_* represents the weight of attention, *s_t_*_−1_ represents the hidden layer state of the DN module at time *t* − 1, s_t_ represents the hidden layer state of the DN module at time *t*, *f* and *g* represent the nonlinear activation function, and *y_t_* represents the predicted output of the DN module at time *t*. *y_t_* is determined by the predicted output *y_t_*_−1_ of the previous moment, the hidden layer state s_t_ of the DN module and the attention semantic coding *C_t_*.

## 3. Implementation Details

All experiments with the text recognition algorithm in this paper are completed in the PyTorch framework. The experimental workstation is equipped with a 3.6 GHz Intel i7-6800k CPU, 64G RAM, eight GTX 2080Ti GPUs, and the operating system is Ubuntu 16.04. In the training process, CUDA 9.0 and Cudnn 7.1 are adopted for GPU acceleration, which can significantly improve the training speed. OpenCV 3.2 with Python 3.6 is used to visualize the results. The parameter settings used in the training process are shown in Table 3.

## 4. Experiments

### 4.1. Experimental Data Set

The data set used in the text recognition algorithm is different from the text detection, which is usually more standard, multilingual and simple. In order to verify the advantages of the text recognition algorithm in this paper, we conducted experimental comparison on a variety of data sets, using the data sets such as SVT, ICDAR 2013, IIIT5K-Words and CUTE80. The following is a detailed introduction. The sample of the scene texts in the data sets in this paper is seen in Figure 7.

SVT [28]: this data set comes from Google Street View. The text size is diverse, the text direction is not fixed, many pictures are polluted by noise and mixed background, and the image resolution is low. This data set can effectively test the text recognition ability of the text recognition algorithm. It contains 647 cropped images and used the two common data formats: SVT-50, SVT-None. “50” means that the annotated dictionary library contains 50 words, and “None” means that there is no dictionary library. The same is true for the following data set.

ICDAR 2013 [29]: this data set is a commonly used data set for text recognition. The text in the image is usually horizontal and the text background is simple. The image format of this data set is the same as that of ICDAR 2003 [30], including 1015 cropped images. The following three data formats are commonly used: ICDAR 2013-50, ICDAR 2013-FULL and ICDAR 2013-None. Each image in this data set has a complete ground truth.

IIIT5K [31]: this data set is collected on the Internet and contains 3000 cropped images. It is a commonly used horizontal text data set. There are three commonly used data formats: III5K-50 with 50 annotated words, III5K-1k with 1000 annotated words, and III5K-None with no annotated words. Each image in this data set has a complete ground truth.

CUTE80 [32]: this data set is a commonly used slanted text or curved text data set, mainly used to evaluate the recognition effect of the algorithm model on multi-direction slanted text and curved text. It contains 288 clipped images and is a data set without dictionary annotation.

### 4.2. Experimental Results and Analysis

In order to verify the effect of the text recognition algorithm and the influence of each sub-module on the text recognition results, the experimental analysis was carried out on each sub-module, and the experimental verification was carried out on the whole text recognition scheme. And in order to validate the importance of each sub-module, we carried on the ablation study [33] in this article. Firstly, we removed the sub-module and test the whole text recognition algorithm, and then added the module and conducted comparison experiment on the whole text recognition algorithm. If there is no significant improvement in the accuracy of text recognition after adding the module, the module will be removed to simplify the algorithm architecture.

As many references use recognition accuracy to evaluate the text recognition algorithm, in order to compare with other text recognition algorithms, this paper adopts recognition accuracy and training time as the evaluation criteria. The following is the detailed experimental results and analysis of the text recognition algorithm.

#### 4.2.1. TPC and its Influence on Text Recognition Results

The TPC module corrects the tilted text into horizontal text through coordinate offset, and uses the EDN module to recognize the horizontal text. In this paper, the importance of the TPC module can be verified by ablation study. The experimental data sets are SVT and IIIT5K, the setting of training parameters is shown in Table 3, the comparison of experimental results is shown in Table 4. In order to demonstrate the effect of TPC module, this experiment selects three images to test the text recognition algorithm. These images all have the characteristics of blur, tilt and bending so as to verify the effect of the text recognition algorithm modified by TPC module.

From the experimental result in Table 4, it can be seen that in the data set SVT the recognition accuracy is significantly improved by more than 6%, which indicates that the text recognition algorithm can more accurately identify the slanted text content after using TPC module to correct the text position. The recognition accuracy in data set IIIT5K is also greatly improved, which indicates that it is also suitable for normal horizontal text. TPC module will increase the training time of the whole model during the training process, but the increase is relatively small and has little impact on the performance.

#### 4.2.2. Dense Connection Network and Its Impact on Text Recognition Results

Dense connection network is an important part of EN module. In order to verify the influence of the network on the whole text recognition algorithm, ablation experiments were carried out. The experimental data sets were ICDAR2013 and IIIT5K. Experimental results are shown in Table 5.

It can be seen from Table 5 that the dense connection network has a great impact on the whole text recognition algorithm and can significantly improve the accuracy of text recognition. In the data sets ICDAR2013 and IIIT5K, with or without dictionary annotation, the accuracy of text recognition is improved by more than 7%, and even reaches 99.4% in IIIT5K-50, which indicates that dense connection network can effectively improve the recognition effect of the text recognition algorithm. After adding the dense connection network, the training time of the model increases little, only about 0.2 h, the result indicates that the dense connection network can improve the back propagation process of the neural network and has a certain optimization effect on the training process of the whole model.

#### 4.2.3. Depth of BLSTM in EN and Its Influence on the Text Recognition Results

This experiment verifies the influence of different depth of BLSTM on the text recognition results. The depth of BLSTM may affect the feature learning ability of the text recognition algorithm. A certain depth can be used to learn more sequence features, but the continuous increase of depth will increase the amount of parameter calculation, result in a longer training time. Therefore, through the experimental comparison of BLSTM with different depth, it is necessary to select the appropriate depth from the accuracy and training time. The experimental data sets are ICDAR2013 and IIIT5K. For the convenience of comparison, the training time is the average time of the text recognition algorithm in three different structures of each data set, and the unit is hour. The setting of training parameters is shown in Table 3, the experimental results are shown in Table 6.

It can be seen from Table 6 that in ICDAR2013-50 the recognition accuracy of BLSTM in the first layer was improved by 7.6% when compared with BLSTM with without BLSTM, which indicates that BLSTM can improve the recognition accuracy of the text recognition algorithm. In addition, as the number of BLSTM layers increases, the recognition accuracy of the algorithm gradually increases and the training time also increases. When the number of BLSTM layers is 2, the recognition accuracy of the text recognition algorithm reaches the highest, which reaches 98.6%, 97.5% and 92.3% in three data formats of ICDAR2013, and reaches 99.4%, 98.1% and 88.3% in three data formats of IIIT5K, respectively. When the number of layers of BLSTM is increased, the recognition accuracy does not increase, but decreases. It is analyzed that the algorithm is too complex, which leads to the over-fitting phenomenon in the nonlinear learning process. At the same time, the number of parameters of the algorithm increases and the training time becomes longer, which is a great challenge to the hardware equipment. Therefore, the two-layer BLSTM is the most reasonable choice in this paper.

#### 4.2.4. Attention Mechanism in DN and Its Influence on Text Recognition Results

Adding attention mechanism to DN can make use of feature information reasonably and improve the decoding efficiency of feature effectively. The effect of attention mechanism on text recognition algorithm was verified by ablation study. The experimental data sets are SVT and IIIT5K, and the setting of training parameters is shown in Table 3, the experimental results are shown in Table 7.

It can be seen from Table 7 that the recognition accuracy with attention mechanism in SVT and IIIT5K can both improved by more than 3%. It shows that the attention mechanism can extract the character effectively in the text recognition algorithm, which is very important to improve the effect of text recognition. At the same time, the training time of the whole model increases by only 0.2 h after the attention mechanism is added, which indicates that the model complexity of the attention mechanism is low and the parameter calculation amount of the whole model does not increase significantly.

### 4.3. Results Compared with Other Text Recognition Algorithms

The above experiments are conducted on the current algorithms for text recognition, and prove the validity of our algorithm in this paper. To validate the effect on tilted text recognition of our text recognition algorithm, we select SVT and CUTE80 as the experimental data sets. The training parameters are shown in Table 3, the experimental results are shown in Table 8.

From the experimental data in Table 8, it can be seen that the text recognition algorithm in this paper can achieve a good recognition accuracy even in the data sets annotated by different dictionaries, no matter the text is tilted or curved. In the two data formats SVT-50 and SVT-None, the text recognition accuracy reached 96.5% and 83.7% respectively. In the curved text data set CUTE80, the recognition accuracy reached 71.3%, indicating that the text recognition algorithm designed in this paper has good robustness for the recognition of slanted and curved text. Because some algorithms do not carry out text recognition experiments in the specified data set, there is represented by “-” in Table 8.

### 4.4. Experimental Results of Text Recognition in the Scene Image of Visual Sensors

For the overall scheme of text recognition in natural scenes captured by visual sensors, the text recognition algorithm is combined with the text detection algorithm for experiments, and a demonstration is designed to directly identify the text images in natural scenes. Figure 8 shows the results.

The test images in Figure 8 were all randomly collected in the natural scene by visual sensors. The image background is complex, the text size is variable, and the text direction is skewed. The red box in the left column is the detection result of the text detection algorithm, while the right column is the recognition result of the text recognition algorithm. According to the recognition results in the right column of Figure 8, the natural scene text recognition algorithm designed in this paper can accurately identify the text in the figure, and has a good recognition effect for both multi-scale text and inclined text, indicating that the algorithm is feasible.

## 5. Conclusions

In this paper, a text recognition algorithm based on TPC-EDN is proposed to solve the problem that text recognition algorithm only with encoding and decoding network is not good for oblique text recognition. Firstly, we analyze the problems existing in the current text position correction method, put forward the TPC method in this paper. Secondly, we consider the text sequence feature, design a text recognition algorithm based on EDN, and adopt the attention mechanism to effectively improve decoding accuracy. Finally, we test the text recognition algorithm and conduct ablation study, compare some experimental data and verify the advantage of our algorithm.

## Figures and Tables

**Figure 1 sensors-20-02942-f001:**
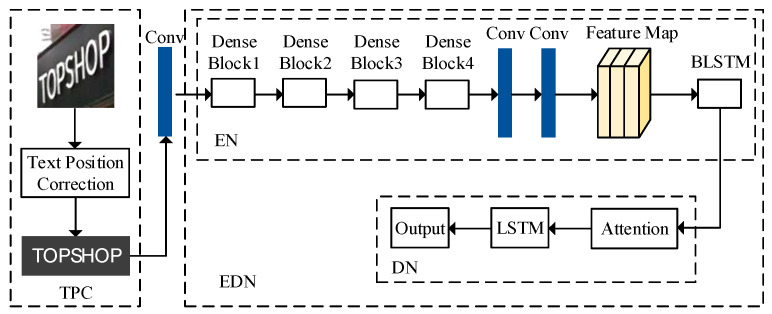
Overall structure of our text recognition algorithm.

**Figure 2 sensors-20-02942-f002:**
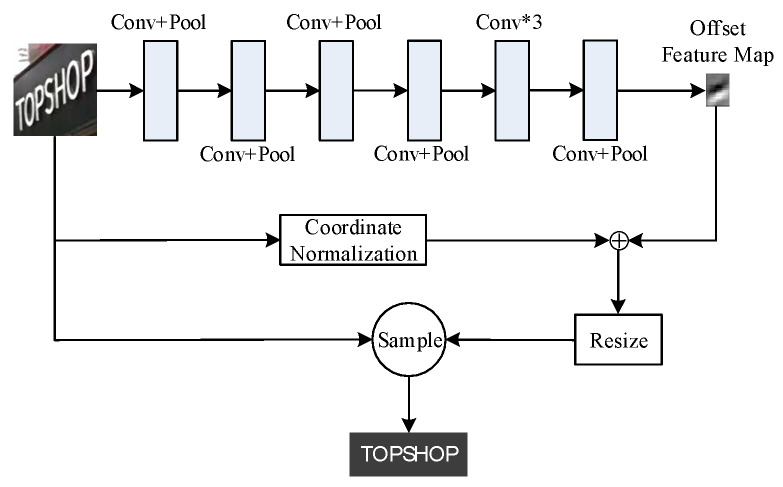
TPC structure diagram.

**Figure 3 sensors-20-02942-f003:**
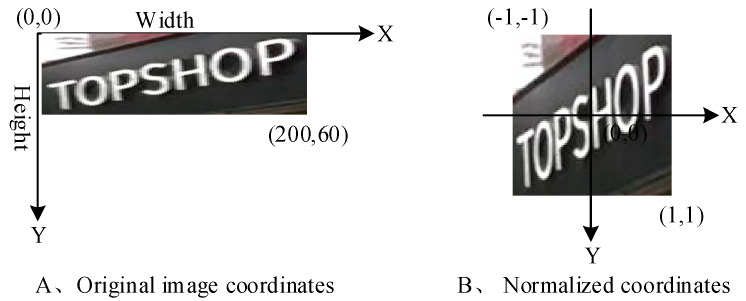
Schematic diagram of coordinate normalization.

**Figure 4 sensors-20-02942-f004:**
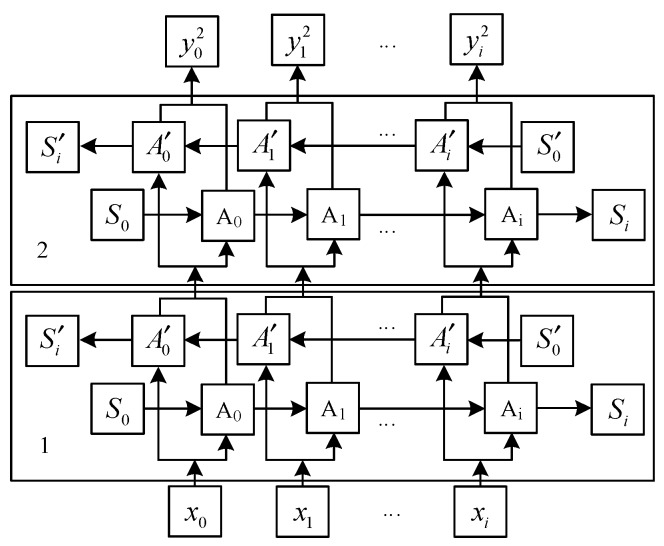
BLSTM structure diagram of two floors.

**Figure 5 sensors-20-02942-f005:**
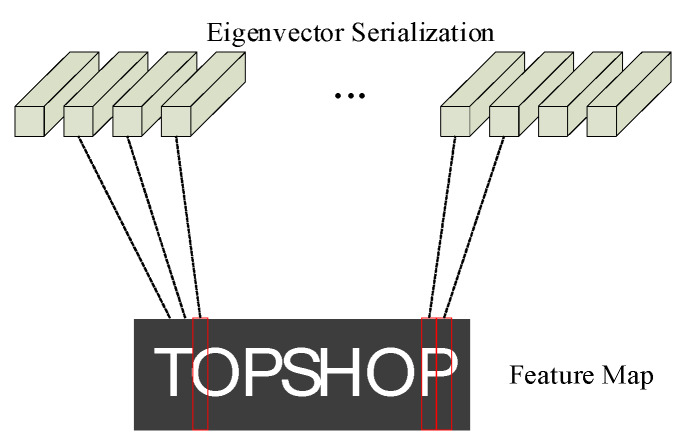
Feature map is transformed into feature vector sequence.

**Figure 6 sensors-20-02942-f006:**
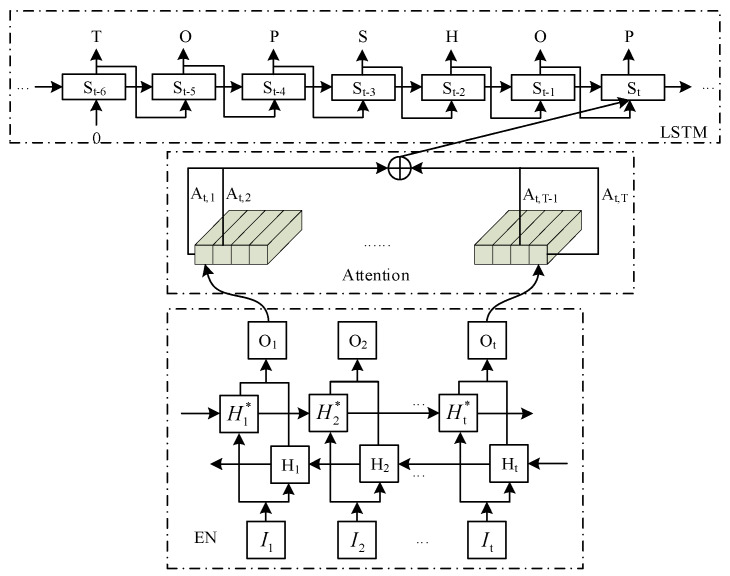
Detail structure on Decoder Network.

**Figure 7 sensors-20-02942-f007:**
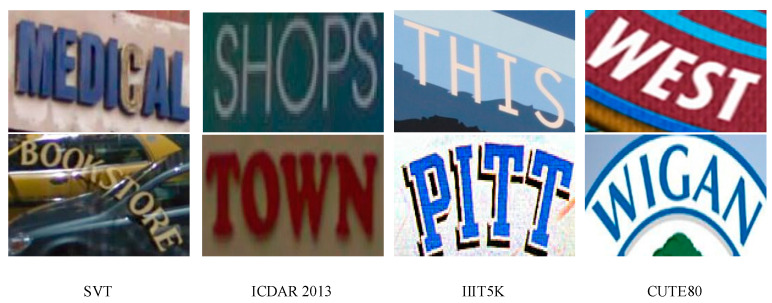
Sample of the scene texts in the data sets in this paper.

**Figure 8 sensors-20-02942-f008:**
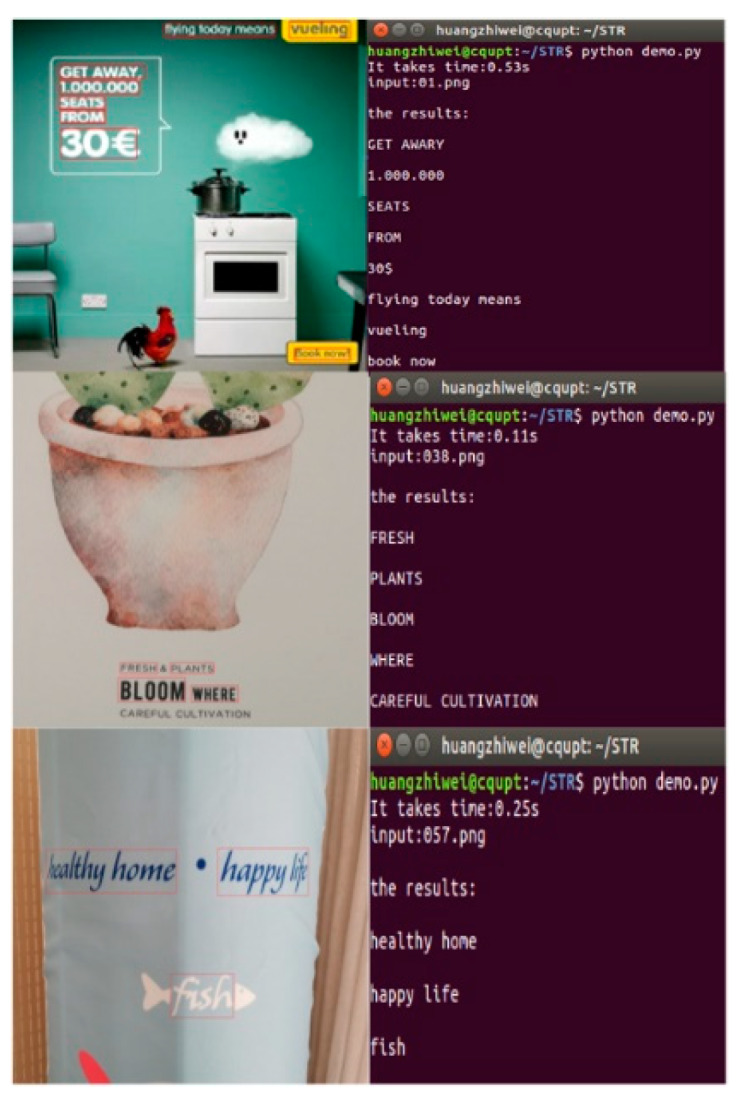
Experimental results of text recognition in a visual sensor scene image.

**Table 1 sensors-20-02942-t001:** Detailed structure of TPC module.

Type	Configuration	Size
The Input	-	1 × 64 ×200
Conv	k3, num64, s1, p1	64 × 64 × 200
AVGPool	k2, s2	64 × 32 × 100
Conv	k3, num128, s1, p1	128 × 32 × 100
AVGPool	k2, s2	128 × 16 × 50
Conv	k3, num256, s1, p1	256 × 16 × 50
AVGPool	k2, s2	256 × 8 × 25
Conv	k3, num128, s1, p1	128 × 8 × 25
AVGPool	k2, s1	128 × 7 × 24
Conv	k3, num64, s1, p1	64 × 3 × 12
Conv	k3, num32, s1, p1	32 × 3 × 12
Conv	k3, num8, s1, p1	8 × 3 × 12
Conv	k3, num2, s1, p1	2 × 3 × 12
AVGPool	k2, s1	2 × 2 × 11
Tanh	-	2 ×2 × 11
The Resize	-	2 × 64 × 200

**Table 2 sensors-20-02942-t002:** Detailed parameters of EN module.

Type	Configuration
Convolution	[k3, num32, s1, p1]
Max Pooling	[k2, s1]
The Activation Function	Swish
Dense Block	[k3, num32, s1, p1] x 4
Dense Block	[k3, num64, s1, p1] x 4
Dense Block	[k3, num128, s1, p1] x 4
Dense Block	[k3, num256, s1, p1] x 4
Convolution	[k3, num128, s1, p1]
Max Pooling	[k2, s1]
The Activation Function	Swish
Convolution	[k3, num128, s1, p1]
BLSTM	Hidden unit: 256
BLSTM	Hidden unit: 256

**Table 3 sensors-20-02942-t003:** Parameter settings

Type	Configuration
The input size	64 × 200
The iterations	100,000
Batch size	16
Learning rate	10^−3^
Learning rate attenuation	0.9/10,000 of the iterations

**Table 4 sensors-20-02942-t004:** TPC’s recognition accuracies (%) and its ablation study.

Model		SVT			IIIT5K		
	50	None	Time	50	1 k	None	Time
Without TPC	89.6	76.5	3.2 h.	93.2	92.5	81.6	6.1 h.
With TPC	96.5	83.7	3.6 h.	99.4	98.1	88.3	6.7 h.

**Table 5 sensors-20-02942-t005:** Recognition accuracies (%) of dense connection network and its ablation study.

Model		ICDAR 2013				IIIT5K		
	50	FULL	None	Time	50	1 k	None	Time
Without DCN	89.6	86.5	79.5	5.4 h.	91.2	89.5	80.5	6.5 h.
With DCN	98.6	97.5	92.3	5.5 h.	99.4	98.1	88.3	6.7 h.

**Table 6 sensors-20-02942-t006:** Depth of BLSTM in EN and its recognition accuracies (%).

Depth		ICDAR 2013				IIIT5K		
	50	FULL	None	Time	50	1 k	None	Time
0	89.6	89.5	83.5	3.4 h.	91.2	90.5	82.6	4.5 h.
1 layer	97.2	94.6	89.9	4.7 h.	93.6	93.3	85.1	5.6 h.
2 layers	98.6	97.5	92.3	5.5 h.	99.4	98.1	88.3	6.7 h.
3 layers	98.3	97.1	92.2	6.8 h.	99.3	97.8	88.1	7.9 h.

**Table 7 sensors-20-02942-t007:** Attention mechanism and its recognition accuracies (%).

Model		SVT			IIIT5K		
	50	None	Time	50	1 k	None	Time
Without Attention	92.7	80.5	3.4 h.	95.5	93.5	84.6	6.5 h.
With Attention	96.5	83.7	3.6 h.	99.4	98.1	88.3	6.7 h.

**Table 8 sensors-20-02942-t008:** Recognition accuracies (%) compared with other text recognition algorithms.

Model	SVT-50	SVT-None	CUTE80
Bissacco et al. [34]	90.4	78.0	-
He et al. [35]	95.4	80.7	-
Jaderberg et al. [36]	93.2	71.7	42.7
Lee et al. [37]	96.3	80.7	-
Shi et al. [38]	96.4	80.8	54.9
Shi et al. [39]	95.5	81.9	59.2
Yang et al. [40]	95.2	-	69.3
Ours	96.5	83.7	71.3

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
