# Peer review of "An Algorithm Based on Text Position Correction and Encoder-Decoder Network for Text Recognition in the Scene Image of Visual Sensors"

_sensors, 2020, doi:10.3390/s20102942_

Round 1
Reviewer 1 Report
The authors presented an interesting approach to text recognition in images. The proposed approach is well presented as well as the results, which are competitive when compared to previous works.
To improve the work, the introduction must be improved, I would suggest the following discussions:
- Introduce the research topic within the research area
- Contextualization
- Progress in the area
- Unresolved conflict or problem
- Limitations of previous works
- Specify the purpose of the work -- main goals
- Present a novel approach, method, or technique
- Describe the proposed methodology
- List of main contributions
There is no previous works or related works section. Therefore, the introduction have to present previous conflicts, problems and limitations as well as approaches proposed to overcome such issues/limitations.
Section 2.2 title is not following the word/latex template?
In the dataset description, the authors could say if the images contains rotated texts, because several readers can not kwon those datasets.
Table 4 title is alone at the bottom of the page.
Are the values of accuracy in Table 4 described in percentage?
Is the resulting image good enough to be used by an OCR algorithm?
For instance, the example of Figure 7 could be with rotated texts.
Other:
In introduction, citation after period: “…such as arcs. [13]”
Author Response
Major concerns:
To improve the work, the introduction must be improved, I would suggest the following discussions:
- Introduce the research topic within the research area
- Contextualization
- Progress in the area
- Unresolved conflict or problem
- Limitations of previous works
- Specify the purpose of the work -- main goals
- Present a novel approach, method, or technique
- Describe the proposed methodology
- List of main contributions
There is no previous works or related works section. Therefore, the introduction have to present previous conflicts, problems and limitations as well as approaches proposed to overcome such issues/limitations.
Ans: Thanks for the reviewer's good suggestion on the introduction. According to your suggests, we have modified and improved the Introduction section item by item so that the manuscript is designed to be more reasonable and the method is described more completely. Each item has been addressed carefully in our revised manuscript, and the revised part is highlighted in red in re-submitted version.
Section 2.2 title is not following the word/latex template?
Table 4 title is alone at the bottom of the page.
In introduction, citation after period: “…such as arcs. [13]”
Ans: We are grateful for these comments. We have corrected these errors accordingly.
In the dataset description, the authors could say if the images contain rotated texts, because several readers can not know those datasets.
Ans: Thank you for pointing out. We have added Figure 7 as a sample of the rotated texts in the database.
Are the values of accuracy in Table 4 described in percentage?
Ans: Yes. We have added a clear description of the units in the Table 4 to avoid misunderstanding.
Is the resulting image good enough to be used by an OCR algorithm?
For instance, the example of Figure 7 could be with rotated texts.
Ans: Yes. We have replaced one and added a text recognition image in Figure 8 to prove that our algorithm can recognize rotation text effectively.
Each comment has been addressed carefully in our revised manuscript, and the revised part is highlighted in red in re-submitted version.

Reviewer 2 Report
This paper presents a scene text recognition technique for scene texts with various geometric distortions such as perspective distortion. The technique consists of two modules, namely, a text position correction (TPC) module that aims to remove geometric distortion to restore text instances to be horizontal and an encoder-decoder network (EDN) that aims to recognize the restored scene text. Experiments show promising results.
I have a few concerns on this paper. For the method part, the technical novelty is incremental and the whole system is more like an integration of existing techniques. Specifically, the TPC module leverages largely deformable convolution network and migration sampling in [21] and [22]. More importantly, this part lacks details on how rectification is achieved, e.g. what are training data, what are loss functions, etc. As far as I know, the spatial transformer network (SPM) [1] could handle such geometric correction. I’d like to know how the proposed TPC performs with respect to the SPM. For EDN, the encoder network is based on dense connection network in DenseNet [26] and BLSTM, and the decoder network is based on the soft attention mechanism [27]. I don’t see much innovation in this part either.
For the experiments, the quantitative results are quite confusing. One major problem is that the comparison and ablation studies keep jumping among different datasets. Besides, qualitative experimental results are missing. For example, the illustration of the TPC results including success and failure cases will be very useful.
Another issue is about literature review which missed quite a number of relevant works. For geometric correction, there are quite a few recent works [2, 3] on this task but focus on more challenging curved text line problem. For scene text recognition, the idea of using RNN and LSTM has been reported in one pioneer work [4] and later extended in [5].
Further, this paper contains lots of grammar errors and typos. It requires a thorough check for correction and improvement.
[1] M. Jaderberg, et al, Spatial Transformer Networks, NIPS, 2015.
[2] F. Zhan, et al, ESIR: End-to-End Scene Text Recognition via Iterative Image Rectification, CVPR, 2019.
[3] B. Shi, et al, Aster: An attentional scene text recognizer with flexible rectification. TPAMI, 2018.
[4] B. Su, et al, Accurate scene text recognition based on recurrent neural network, ACCV, 2014.
[5] B. Su, et al, Accurate recognition of words in scenes without character segmentation using recurrent neural network, PR, 2017.
Author Response
Comments and Suggestions for Authors
This paper presents a scene text recognition technique for scene texts with various geometric distortions such as perspective distortion. The technique consists of two modules, namely, a text position correction (TPC) module that aims to remove geometric distortion to restore text instances to be horizontal and an encoder-decoder network (EDN) that aims to recognize the restored scene text. Experiments show promising results.
Author’s Reply:
First and foremost, we would like to express our sincere thanks for your supportable and constructive comments, which helps us make deeper understanding about scene text recognition and further improve the manuscript. According to your comments, we have tried our best to address each comment carefully, and the revised parts are highlighted in red in the re-submitted manuscript. Now, we divide the whole comments given by dear Reviewer#2 to provide point-by-point response to each comment.
Major concerns:
(1) I have a few concerns on this paper. For the method part, the technical novelty is incremental and the whole system is more like an integration of existing techniques. Specifically, the TPC module leverages largely deformable convolution network and migration sampling in [21] and [22]. More importantly, this part lacks details on how rectification is achieved, e.g. what are training data, what are loss functions, etc. As far as I know, the spatial transformer network (SPM) [1] could handle such geometric correction. I’d like to know how the proposed TPC performs with respect to the SPM.
Ans: Thanks for the reviewer's suggests. We propose an improved TPC method based on the idea of variable convolution two-dimensional offset and offset sampling. Specifically, we design a deformable convolution network module in the feature extraction process, which can effectively extract the features of some deformable characters in natural scenes. The network module adopts parallel convolution method to extract multi-scale features of the same layer, and fuses deform convolution to extract text features of different scales and shapes. The network module is divided into four parallel branches, each of which uses a 3×3 deformable convolution kernel, the leftmost branch adds a Max-pooling, and the middle two branches use a 1×1 conventional convolution kernel on the first floor, which is used to reduce the calculation amount of parameters and keep the same dimension of each layer for the convenience of data fusion.
Deformable convolution is an offset learning method on receptive field. By adding two-dimensional offsets to the regular sampling location of the fixed receptive field, the sampling grid can be shaped freely to enhance the expression of features. Moreover, Deformable convolution inherits the spatial invariable property of the conventional convolution, so it is suitable for feature extraction of natural scene texts.
Spatial Transformer Network was a learnable module, the, which explicitly allows the spatial manipulation of data within the network. This differentiable module can be inserted into existing convolutional architectures, giving neural networks the ability to actively spatially transform feature maps, conditional on the feature map itself, without any extra training supervision or modification to the optimization process. The use of spatial transformers results in models which learn invariance to translation, scale, rotation and more generic warping, resulting in state-of-the-art performance on several benchmarks, and for a number of classes of transformations.
Convolutional Neural Networks define an exceptionally powerful class of models, but are still limited by the lack of ability to be spatially invariant to the input data in a computationally and parameter efficient manner. In this paper, our proposed TPC is able to modify the slanted text to horizontal text like Spatial Transformer Network to complete end-to-end training, but in contrast our training process is simple and fast.
(2) For EDN, the encoder network is based on dense connection network in DenseNet [26] and BLSTM, and the decoder network is based on the soft attention mechanism [27]. I don’t see much innovation in this part either.
Ans: Thank you for your comment. Although the partial modules of proposed method are built upon the previous works, there is currently no model integrates those independent part into an end-to-end framework to achieve accuracy and satisfactory text recognition results. It has been noted that we did not utilize the whole framework but the partial module for high-quality text recognition results. For a better representation, we have highlighted these modules and explained the reason why they work well in our situation. The revised part is highlighted in red in re-submitted version.
(3) For the experiments, the quantitative results are quite confusing. One major problem is that the comparison and ablation studies keep jumping among different datasets. Besides, qualitative experimental results are missing. For example, the illustration of the TPC results including success and failure cases will be very useful.
Ans: Thank you for pointing out. I did not notice this until this constructive comment. We have double-checked our source code, and we promise that the experiments are real and objective. After comprehensive analysis, we conclude the different among these datasets is main reason for the jumping results. The training set requires paired image and text description. The higher is the matching degree between image and text, the better are the result. Besides, the larger is the amount of paired image and text, the better are the results.
(4) Another issue is about literature review which missed quite a number of relevant works. For geometric correction, there are quite a few recent works [2, 3] on this task but focus on more challenging curved text line problem. For scene text recognition, the idea of using RNN and LSTM has been reported in one pioneer work [4] and later extended in [5].
Ans: We are grateful for this comment. According to this comment, we have cited all the references suggested, with some corresponding changes in the text. Due to the similarities in research methods and their applications, we have added more related references on our method and other comparisons to make this manuscript more readable and significant.
(5) Further, this paper contains lots of grammar errors and typos. It requires a thorough check for correction and improvement.
Ans: Thanks for the reviewer's suggests. According to your suggestion, we have read the article carefully and made some modifications. Maybe there is still little grammatical mistakes or partially not be flow. If necessary, we will ask for help of professional English editing service.
Each comment has been addressed carefully in our revised manuscript, and the revised part is highlighted in red in re-submitted version.

Round 2
Reviewer 2 Report
Qualitative illustrations of the proposed text position correction (TPC) are required including success cases and failure cases. Correction of geometric distortions is a challenging task and I do not see sufficient details and illustrations on how it is achieved by the proposed TPC.
Besides, some related scene text recognition work is not reviewed, e.g. Multilingual scene character recognition with co-occurrence of histogram of oriented gradients in PR, 2016.
Author Response
Major concerns:
(1) Qualitative illustrations of the proposed text position correction (TPC) are required including success cases and failure cases.
Author’s Reply:
First and foremost, we would like to express our sincere thanks for your approval on other replies. Then, we also appreciate your supportable and constructive comments, which improve the manuscript and encourage us to make in-depth understanding about text position correction. Thanks again for your effort and patient.
As you mentioned, the qualitative results should contain successful and failed cases. Based on this, we have added some new cases as shown in Figure 1, and the revised part is highlighted in red in our re-submitted version.
|
(a) Input Image |
(b) Image Rectified by TPC |
|
Figure 1. Example results of our TPC rectified images |
|
The left column (a) is the input image from the experimental data set, the right column (b) is the rectified image by our TPC. From Figure 1 we can see that our proposed method can achieve satisfactory results in general cases. However, when the image contrast is not high just like in the third row in Figure 1, the correction effect needs to be improved. We plan to focus on these unusual cases in future work.
(2) Correction of geometric distortions is a challenging task and I do not see sufficient details and illustrations on how it is achieved by the proposed TPC.
Author’s Reply:
I can’t agree more about that correction of geometric distortions is a challenging task. Traditional methods such as affine transformation can work well on text with small tilt angle, but they cannot work well on the text with large tilt angle. For this, we design an improved TPC method based on variable convolution two-dimensional migration and migration sampling, which is a coordinate migration regression method based on CNN. It can be combined with other neural networks to complete end-to-end training, and the training process is simple and fast. The detailed processes can be summarized as followed:
(a)Pre-process is carried out to process the input text into the same size, which can speed up the training process of the algorithm.
(b)The spatial features of pixels are extracted by CNN to obtain a fixed size feature map, in which each pixel corresponds to a part of the original image. This is equivalent to splitting the original image into several small pieces, and the prediction of coordinate offset for each piece is the same as the two-dimensional offset prediction of the deformable convolution.
(c)The offset is then superimposed on the normalized coordinates of the original image.
(d)The Resize module uses a bilinear interpolation method to sample the text feature map to the original size as the revised text.
(3) Besides, some related scene text recognition work is not reviewed, e.g. Multilingual scene character recognition with co-occurrence of histogram of oriented gradients in PR, 2016.
Author’s Reply:
Thanks a lot. We have reviewed some related scene text recognition work.
We would like to express our sincere thanks to dear Reviewer#2 for your comment and effort again. Your comments have made significant contribution to both this manuscript and us. Thank you!
